# Simultaneous Speech and Eating Behavior Recognition Using Data Augmentation and Two-Stage Fine-Tuning

**DOI:** 10.3390/s25051544

**Published:** 2025-03-02

**Authors:** Toshihiro Tsukagoshi, Masafumi Nishida, Masafumi Nishimura

**Affiliations:** 1Graduate School of Science and Technology, Shizuoka University, 3-5-1 Johoku, Chuo-ku, Hamamatsu 432-8011, Japan; tsukagoshi.toshihiro.20@shizuoka.ac.jp (T.T.); nishida@inf.shizuoka.ac.jp (M.N.); 2Department of Smart Design, Faculty of Architecture and Design, Aichi Sangyo University, 12-5 Harayama, Oka-cho, Okazaki 444-0005, Japan

**Keywords:** speech recognition, health monitoring, eating behavior recognition, self-supervised learning, skin-contact microphones

## Abstract

Speaking and eating are essential components of health management. To enable the daily monitoring of these behaviors, systems capable of simultaneously recognizing speech and eating behaviors are required. However, due to the distinct acoustic and contextual characteristics of these two domains, achieving high-precision integrated recognition remains underexplored. In this study, we propose a method that combines data augmentation through synthetic data creation with a two-stage fine-tuning approach tailored to the complexity of domain adaptation. By concatenating speech and eating sounds of varying lengths and sequences, we generated training data that mimic real-world environments where speech and eating behaviors co-exist. Additionally, efficient model adaptation was achieved through two-stage fine-tuning of the self-supervised learning model. The experimental evaluations demonstrate that the proposed method maintains speech recognition accuracy while achieving high detection performance for eating behaviors, with an F1 score of 0.918 for chewing detection and 0.926 for swallowing detection. These results underscore the potential of using voice recognition technology for daily health monitoring.

## 1. Introduction

Speaking and eating are essential components of human health management. Regarding eating behavior, research indicates that the frequency and speed of chewing are associated with an increased risk of obesity and diabetes [1,2,3]. Moreover, thorough chewing followed by swallowing has been shown to suppress postprandial blood glucose spikes, thereby reducing the risk of lifestyle-related diseases [4]. Among elderly individuals, a decline in swallowing function can significantly impact health and daily life. Regular screening for swallowing function is crucial in reducing the risks of aspiration pneumonia and choking [5].

Traditional methods for observing swallowing actions, such as video fluoroscopic (VF) and the fiberoptic endoscopic evaluation of swallowing (FEES) [6], are widely used. However, these methods require specialist intervention, which is costly and involves invasive procedures, posing several challenges.

Non-invasive approaches for recognizing eating behaviors have been proposed, using inertial data and bioacoustic signals [7,8,9]. Methods leveraging inertial data aim to estimate eating behaviors by capturing subtle movements around the throat. For instance, Nguyen et al. [7] proposed a method involving the attachment of inertial measurement units around the neck. By training a long short-term memory network on inertial data obtained from the throat region, they demonstrated the ability to detect chewing and swallowing actions. Similarly, approaches using bioacoustic signals generated near the throat during eating have also been explored. These signals can be collected using skin-contact microphones [8] or earbud-integrated microphones [9]. Notably, skin-contact microphones directly record eating-related sounds near the throat, minimizing external noise and enabling the collection of high-quality sound data related to eating behaviors [10,11].

Conversely, speaking can serve as an important indicator for daily health monitoring. Analyzing speech content has shown potential in facilitating the early detection of conditions such as Alzheimer’s disease [12] and depression [13]. Speech features such as vocabulary size, fluency, and coherence are critical indicators of cognitive decline. To enable such health monitoring, it is essential to develop technology capable of accurately recognizing natural speech in daily life. Because throat microphones exhibit different acoustic characteristics than conventional speech microphones, they inevitably affect speech recognition accuracy. However, self-supervised learning (SSL) is known to be effective in addressing this issue.

Recent advancements in deep learning have significantly improved speech recognition technology. In particular, the introduction of SSL has greatly enhanced recognition accuracy [14,15,16]. Wav2vec 2.0 [14] is a pre-trained model that learns acoustic features from large volumes of unlabeled data, enabling high recognition accuracy, even with limited labeled data. Specifically, XLSR-53 [17], pre-trained in 53 languages, has demonstrated exceptional performance in multilingual speech recognition. Additionally, it has proven effective for adapting to resource-scarce domains and downstream tasks beyond speech recognition [18]. Leveraging the versatility of such pre-trained models, studies employing wav2vec 2.0 features for multitask learning have advanced. Beyond speech recognition tasks, improvements in emotion recognition [19,20], voice activity detection [21], and speaker identification [22] have also been reported.

Connectionist Temporal Classification (CTC) [23], a widely used technique in speech recognition, enables learning when the input and output sequence lengths differ, without requiring explicit alignment. This loss function has been extensively applied to train speech recognition models and has also demonstrated applicability in acoustic event detection tasks [24]. In the domain of eating behavior recognition, combining SSL with CTC loss has yielded high recognition accuracy [25]. However, real-world dining scenarios often present complex acoustic environments where eating sounds and speech occur intermittently in a time series. To address these complexities, researchers have explored integrated recognition models that process eating and speech sounds simultaneously. For instance, a method was proposed that utilizes acoustic features extracted from wav2vec 2.0, which are then input into independent decoders specialized for eating behavior recognition and speech recognition. This multitask learning framework optimizes both tasks concurrently through weighted loss [26]. However, this multitask learning approach faces challenges, particularly domain mismatches between speech and eating sounds, which complicate the optimization of feature representation learning. A specific issue arises in achieving accurate temporal alignment for event detection in the CTC decoder for eating behavior recognition. While SSL-based feature representations have shown effectiveness, new strategies are needed to ensure consistent feature representations across diverse acoustic domains.

We introduce a novel approach for the simultaneous recognition of speech and eating sounds using skin-contact microphones. The approach features a data augmentation method that combines speech and eating sounds and a two-stage fine-tuning procedure designed to facilitate domain adaptation.

This study is structured as follows: Section 2 describes the audio recording methodology using skin-contact microphones and outlines the proposed approach for simultaneous recognition of speech and eating sounds. Section 3 details the experimental setup, including datasets, model configurations, and evaluation metrics. Section 4 presents comprehensive results of our experiments, comparing the performance of various methods across different evaluation frameworks. Section 5 provides an in-depth discussion of our findings, analyzing the efficacy of the proposed data augmentation and two-stage fine-tuning techniques, and interpreting the experimental outcomes. Section 6 summarizes the study’s key contributions, outlines limitations of the current approach, and suggests promising directions for future research in this domain. The outcomes of this study underscore the potential of integrating speech recognition technology into daily health monitoring applications, paving the way for advancement in this field.

## 2. Approach

### 2.1. Skin-Contact Microphones

This study introduces a novel method for audio data collection using skin-contact microphones to accurately capture speech and eating behaviors. Skin-contact microphones, which record sound through direct contact with the skin, offer superior resistance to external noise and are particularly effective for capturing eating behaviors. As illustrated in Figure 1, two types of skin-contact microphones were strategically placed in four distinct locations: beneath the left and right ears and on the upper and lower parts of the throat. These positions were selected to capture both eating sounds and speech. We designed the microphone arrangement under the hypothesis that sensors below the ears might capture vertical jaw movements during chewing—distinguishing sounds from each side of the jaw and detecting subtle acoustic changes from food movement. Similarly, microphones on the upper and lower throat were positioned with the expectation that they could record the movement of food during swallowing, potentially enabling a chronological and spatial depiction of the process from its onset onward. It should be noted that skin-contact microphones tend to also capture blood flow vibrations; to remove the noise arising from these vibrations, a high-pass filter with a cutoff frequency of 100 Hz was applied. Signals from the four skin-contact microphones and one close-talk microphone (a total of five channels) were recorded in parallel at 16-bit/44.1 kHz, and then downsampled to 16 kHz. Figure 2 presents examples of eating behavior sounds recorded using skin-contact microphones. The corresponding spectrogram illustrates 15 chewing events and one swallowing event involving cabbage. Subfigure (e) shows data simultaneously recorded using a close-talk microphone for comparison, highlighting that the close-talk microphone was less effective in capturing eating behavior sounds. Despite the added effort required to mount the skin-contact microphones, they provide valuable insights into potential declines in chewing and swallowing functions, even when used for brief periods during specific meals, such as breakfast. 

### 2.2. Annotation of Eating Behavior Sound Data

Figure 3 presents an example of annotations for below-ear microphone audio. For the recorded eating behavior sound data, strong labels indicating the precise times of each eating behavior event were manually assigned by referencing the simultaneously captured video footage. From these strong labels, the time information was removed, and only the sequence of events was extracted, which was then defined as a weak label.

During model training, weak labels that retain the order of the eating behavior events were used as supervision signals. For evaluation, strong labels were used as ground truth data to verify the temporal accuracy. This labeling approach, which distinguishes between temporal information and event sequences, enables the construction of a dataset that adequately considers the continuity and temporal patterns of eating behaviors, thereby contributing to improved recognition accuracy.

### 2.3. Proposed Method

Figure 4 illustrates the structure of the simultaneous speech and eating behavior recognition model alongside its learning methodology.

#### 2.3.1. Two-Stage Fine-Tuning for Efficient Model Adaptation

In this section, we describe the implementation of a two-stage fine-tuning model adaptation strategy, which is tailored to address differences in task complexity. Speech recognition requires distinguishing thousands of vocabulary items, whereas eating behavior recognition involves identifying only a few types of events. Training both tasks from the start could lead to suboptimal performance, as the model might struggle to balance the high granularity needed for speech recognition with the relatively simpler event detection required for eating behavior recognition. By introducing a two-stage fine-tuning framework, we ensure that the model first establishes robust linguistic representations before incorporating eating behavior recognition, ultimately leading to more stable and effective learning across both tasks.

In the first stage, we adapt the self-supervised pre-trained model to Japanese speech recognition. This stage establishes the fundamental learning required for complex speech recognition. In the second stage, additional training was conducted for eating behavior recognition using the synthetic data generated as described earlier. The proposed method improved the accuracy of eating behavior recognition while maintaining the performance of speech recognition.

The main advantage of this two-stage fine-tuning approach lies in its ability to align with task complexity. Specifically, by first addressing the more complex task of speech recognition tasks, followed by the relatively simple eating behavior recognition, the distinct characteristics of both tasks were effectively captured. This strategy facilitates high-accuracy recognition for combined speech and eating behavior tasks, even with limited training data.

#### 2.3.2. Data Augmentation Using Synthetic Speech and Eating Behavior Data

Developing a simultaneous recognition model for speech and eating behaviors in real-world scenarios poses challenges due to the lack of suitable training data. To mitigate this challenge, synthetic data were generated by combining existing speech corpora and eating behavior sound corpora. This approach replicates natural transitions, such as “speech → eating behavior” and “eating behavior → speech”, thus enriching the training data with transitions between acoustic domains and enhancing model robustness.

In particular, audio files from both domains were listed, and their durations were measured. The synthesis of data involved two methods based on the measured durations:
For audio clips lasting between 7 and 10 s, white noise was appended to both ends, normalizing their duration to a fixed length of 10 s. This preprocessing step ensured a consistent input length for the model.For audio clips shorter than 7 s, two clips were concatenated to create new data. The domain of the concatenated audio (speech or eating behavior) was carefully determined to maintain a balanced ratio of speech and eating sounds within the dataset. To fill any remaining duration, additional audio was randomly selected. White noise was then added before, between, and after the clips to ensure uniformity in the dataset.

#### 2.3.3. Model Architecture

This study introduces a model designed to simultaneously recognize speech and eating behavior sounds. The model integrates wav2vec 2.0, followed by four fully connected layers, and employs a CTC loss for training. This architecture harnesses pre-trained feature representations to output transcriptions for speech recognition and event labels, “chewing” and “swallowing”, for eating behavior recognition. Using the same framework allows the model to identify acoustic events across different domains. The detailed components of the model are discussed below.

Wav2vec 2.0 [14] is an SSL model extensively used in speech recognition and event detection. The proposed method directly processes normalized raw audio waveforms as input, extracting latent features that capture not only linguistic characteristics (such as those found in speech) but also a broad range of acoustic properties. The architecture features a seven-layer convolutional neural network for feature extraction, followed by discretization for contrastive learning. During training, parts of the latent representations are masked and input into a transformer. The model learns by minimizing the loss between the outputs of masked regions and original latent representations, thus yielding adaptable audio features suitable for various tasks. Fine-tuning wav2vec 2.0 has demonstrated high-accuracy speech recognition capabilities, even with relatively small datasets.

CTC facilitates the alignment between the audio time frame and sequential character outputs. By introducing a blank symbol (_), CTC addresses the mismatch between the input and output sequence lengths. This flexibility makes it suitable for tasks beyond speech recognition, such as identifying eating behaviors from meal sounds. CTC efficiently pinpoints timestamps for specific acoustic events in time-series data.

For implementation, this study employed the open-source SpeechBrain framework [27]. SpeechBrain offers specialized functionalities for tasks like speech recognition and speaker identification, including support for wav2vec 2.0 and CTC loss. This framework enabled efficient training and evaluation of the proposed model.

### 2.4. Comparative Methods

To evaluate the performance of the integrated speech and eating behavior recognition model, multiple learning methods were implemented, and comparative experiments were conducted. The following experiments were designed to independently verify the effectiveness of the two-stage fine-tuning approach and the impact of synthetic data. Figure 5 presents an overview of these comparative methods.

As baseline single-task models (A) for performance evaluation, two single-task models were trained: (1) Speech recognition model (A1): trained exclusively on speech data during the first stage of learning. (2) Eating behavior recognition model (A2): trained exclusively on eating sound data during the first stage of learning. Both models were developed based on the simultaneous speech-eating behavior recognition framework illustrated in Figure 4. These baselines serve as reference points for further evaluation. Two approaches using single-stage fine-tuning were evaluated: (1) independent datasets (B1): This approach treated speech and eating behavior datasets as independent training data and performed recognition using the same model. (2) Synthetic data (B2): This approach used synthetic data generated by the method described in Section 2.3.2 to enhance recognition accuracy. These comparisons were instrumental in assessing the contribution of the synthetic data to the model’s performance.

Furthermore, for the two-stage fine-tuning approach (C), we first trained the model on the speech recognition task during the first stage, followed by fine-tuning with different datasets in the second stage. We evaluated three different methods for the second-stage fine-tuning to assess their effectiveness. The first method (C1) involves exclusively eating behavior sound data. This approach performs additional training using data specifically tailored for eating behavior recognition on a model previously trained for speech recognition. The second method (C2) incorporates both speech and eating behavior sound datasets for fine-tuning. This method aims to enhance eating behavior sound recognition capabilities while retaining the model’s speech recognition performance. Finally, the third method (C3) uses synthetic data for fine-tuning. Here, synthetic datasets were used during the second stage to investigate their potential impact on performance. These comparisons facilitate a stepwise evaluation of the two-stage fine-tuning approach (C) and the contribution of synthetic data (methods B2 and C3) to the overall model performance. The main objective of implementing the speech and eating behavior recognition tasks within a unified model is to evaluate how closely the model’s performance aligns with the baseline benchmarks for each task individually.

## 3. Experiments

### 3.1. Training Datasets—Speech

We used Common Voice Dataset 11.0 (hereafter CommonVoice11) as the speech dataset. CommonVoice [28] is a crowdsourcing project for speech data, where volunteers worldwide contribute by recording and verifying their speech to build a large-scale open corpus. In this study, we focused specifically on verified Japanese speech data and excluded files exceeding 10 s in length. This decision ensured consistency with the input formats for eating sound data and improved learning efficiency. As a result, the speech training dataset comprised 15,455 utterances, corresponding to approximately 53 h of speech data.

To enhance the dataset, we employed several data augmentation techniques:Speed modulation [29]: The sampling rate of the original waveform is adjusted to alter the speed without distorting the pitch.Time-domain dropout [30]: This method replaces random time intervals in the waveform with zeros.Frequency-domain dropout [31]: This masks specific frequency bands in the audio signal.

These augmentation methods were also applied to the eating behavior sound data. The datasets were divided into training and validation sets with a 9:1 split, facilitating model training and performance evaluation.

### 3.2. Training Datasets—Eating Behavior Sounds

The training dataset for eating behavior sounds was collected from 27 participants. The dataset includes the recordings of 17,539 chewing and 1329 swallowing events. To ensure diverse data, we recorded sounds of foods with varying textures: cabbage, crackers, gum, and water. For solid foods, both chewing and swallowing sounds were recorded, whereas only the swallowing sounds were targeted for water. Following previous research [24], the signals from the four-channel microphones were integrated into a single input by additional processing. This integration allowed for efficient learning with a unified model while leveraging the distinct characteristics of each microphone position. Specifically, below-ear microphones were optimized for detecting chewing sounds, and throat-position microphones for swallowing sounds.

The eating behavior sound data often feature consecutively identical labels, with chewing events dominating. To address this, we adopted the label-initiated segment augmentation (LISA) technique proposed in a previous study [24]. LISA generates up to 10 s starting from the beginning of each label, maintaining temporal continuity while augmenting the dataset. We applied the same acoustic data augmentation methods (speed modulation, time-domain dropout, and frequency-domain dropout) to eating behavior sound data. The data were then split into training and validation sets with a 9:1 ratio, supporting both model training and evaluation.

### 3.3. Training Datasets—Generation of Synthetic Data

Based on the speech dataset (CommonVoice11) described in Section 3.1 and the eating behavior sound dataset described in Section 3.2, we generated synthetic data using the method described in Section 2.3.2. This resulted in a synthetic dataset comprising 35,177 utterances, totaling approximately 98 h of data. The generated dataset was partitioned into training and validation subsets at a 9:1 ratio, facilitating model training and performance evaluation.

### 3.4. Test Dataset

For model evaluation, speech and eating sound data were collected from five subjects who were excluded from the training dataset. For speech data, we utilized the newspaper article reading corpus [32], which was designed to support the development of Japanese large-vocabulary continuous speech recognition systems. Recordings were conducted in a quiet laboratory environment, with each subject producing 40 utterances captured through four contact microphone channels. For eating sounds, data from five subjects were utilized, including 3800 chewing events and 270 swallowing events. During the evaluation, the same signal processing method employed during training was applied using the sum of the signals from the four microphone channels.

### 3.5. Evaluation Metrics

We used multiple metrics to evaluate model performance. For speech recognition, the character error rate (CER) was employed. For eating sound recognition, F1 scores were calculated for each class (chewing and swallowing). The F1 score calculation was based on strong labels containing temporal information, as described in Section 2.2. Since strong labels specify the timing of each eating event, the temporal consistency between model outputs and actual events could be assessed. CTC typically generates peak-like outputs over very short intervals of 1–4 frames (1 frame = 20 milliseconds). To account for this characteristic, we considered a prediction correct if the CTC output overlaps with the interval of eating events indicated by strong labels.

Additionally, allowances were set before and after strong label intervals to accommodate the unique characteristics of each event. Specifically, for swallowing, due to the critical importance of accurate timing for medical applications, such as aspiration risk assessment, a strict allowance of 0.01 s was set before and after the strong label interval. For chewing, as frequency estimation is the main evaluation metric, an allowance of 0.05 s was applied before and after the strong label interval. These tailored evaluation criteria enable assessments that are aligned with the specific characteristics and practical requirements of each event.

### 3.6. Model Training Configuration

We utilized wav2vec2-large-xlsr-53 [17] as the pre-trained model, keeping the feature extractor parameters fixed while updating the remaining parameters. The model architecture included four fully connected (FC) layers appended to the Wav2vec2 output layer, each with an output dimension of 1024. Each FC layer incorporated 1024 neurons, LeakyReLU activation functions, batch normalization, and dropout (probability 0.15) to mitigate overfitting.

For optimization, AdaDelta [33] (learning rate: 1.0, ρ = 0.95, ε = 1 × 10^−8^) was applied to the main model, while Adam [34] (learning rate: 0.0001) was used for the Wav2vec2 component. Learning rates were reduced by factors of 0.8 for the main model and 0.9 for the Wav2vec2 component when the relative improvement in validation loss was less than 0.25% (0.0025).

CTC decoding employed a beam search algorithm with a beam width of 100. The decoding parameters included a log probability pruning threshold of −12.0 and a minimum log probability for token pruning of −1.2, with history pruning enabled. Language model score integration was not implemented. The final model was selected based on the checkpoint with the minimum loss value in the validation set.

## 4. Results

In this experiment, we compared multiple training methods to achieve simultaneous recognition of speech and eating sounds. The evaluation results presented in Table 1 confirm the effectiveness of the proposed method, which combines two-stage fine-tuning and synthetic data. Here, we describe the detailed evaluation results for each method.

### 4.1. Baseline Model Performance Evaluation

As single-task baseline models, we evaluated the speech recognition model (A1) and the eating sound recognition model (A2). The speech recognition model (A1) achieved a CER of 15.19% on the test set. Meanwhile, the eating sound recognition model (A2) demonstrated high recognition accuracy, with F1 scores of 0.905 for chewing events and 0.963 for swallowing events. These results represent the baseline performance when each task was performed independently.

### 4.2. Evaluation of Single-Stage Fine-Tuning

The method of treating speech and eating sounds as separate training datasets (B1) achieved a CER of 17.13% for speech recognition and F1 scores of 0.769 for chewing and 0.961 for swallowing. While maintaining relatively high performance for swallowing, this method showed significant degradation in terms of chewing and speech recognition accuracy. In contrast, the method using synthetic data (B2) improved the CER to 16.38%, showing better performance in speech recognition compared to B1. However, the eating sound recognition performance was insufficient, with F1 scores decreasing to 0.835 for chewing and 0.654 for swallowing.

### 4.3. Evaluation of the Two-Stage Fine-Tuning

The two-stage fine-tuning approach using only eating behavioral sound data (C1) resulted in a significant deterioration in speech recognition performance, with a CER of 87.45%. Conversely, the method using both speech and eating behavioral sound data (C2) maintained speech recognition with a CER of 16.22%; however, the eating behavior recognition performance declined, resulting in a chewing F1 score of 0.243. In contrast, the second-stage fine-tuning approach using synthetic data (C3) achieved high detection accuracy, with F1 scores of 0.918 for chewing and 0.926 for swallowing in eating behavior recognition, while relatively maintaining the baseline standard with a speech recognition CER of 16.24%.

## 5. Discussion

### 5.1. Effect of Synthetic Data Use

The proposed learning approach using synthetic data augmentation proved effective for the simultaneous recognition of speech and eating behavioral sounds. Compared to training with independent data (B1), training with synthetic data (B2) significantly improved the F1 score for chewing detection. This performance improvement suggests that the model effectively learned the timing information of eating behavioral sounds occurring at various positions within speech by synthesizing speech and eating behavioral sounds of varying lengths and combinations. These results demonstrate that training with data synthesis is effective for acoustic feature separation and recognition of speech and eating behaviors.

### 5.2. Effect of Two-Stage Fine-Tuning

The proposed two-stage fine-tuning approach revealed critical insights into the balance between linguistic and behavioral feature representations. Excessive overwriting of the linguistic feature representations acquired during the first stage of learning occurred through additional training with eating behaviors alone (C1), resulting in a significant deterioration in speech recognition performance (CER 87.45%). These findings suggest that excessive adaptation to a specific domain compromises the model’s linguistic acoustic feature representations.

On the other hand, the method of simultaneously learning speech and eating behavioral sounds (C2) resulted in overfitting the temporal feature representations of speech recognition, which led to difficulties in capturing appropriate temporal positions in eating behavior detection. Performance improvement was observed when combined with synthetic data augmentation (C3). This improvement is likely attributable to the acquisition of more flexible feature representations through a stepwise approach: obtaining basic speech recognition capabilities using large-scale speech data before learning eating behavior recognition. Additionally, incorporating various temporal combination patterns of speech and eating behavioral sounds into the training data through synthetic data likely facilitated more accurate temporal information representations in both domains.

As shown in Figure 6, the improvement in feature representation via the proposed method is evident. For inputs containing both speech and eating behaviors, the baseline model (A2) incorrectly detected speech segments as chewing events. This indicates that the model trained only on eating behaviors misrecognized acoustic variations in speech as eating behaviors. In contrast, the proposed method (C3) clearly distinguishes between speech segments and eating behavior events and produces appropriate labels. These results suggest that through two-stage fine-tuning and the use of synthetic data, the model acquired feature representations capable of appropriately identifying and separating the acoustic characteristics of both speech and eating behaviors. The suppression of false detections in speech is a particularly important characteristic for practical application.

### 5.3. Limitations and Future Directions

This study has certain limitations arising from the nature of the evaluation data. The test data were recorded in a laboratory environment and did not fully reflect the diversity of food types encountered in actual eating scenarios. Additionally, future research should address more complex acoustic event interactions, such as natural overlap patterns between speech and eating behaviors. Expanding the evaluation to real-world settings and incorporating a broader range of eating contexts will enhance the robustness and applicability of the proposed method.

## 6. Conclusions

In this study, we proposed a learning method that combines synthetic data augmentation with a two-stage fine-tuning approach to simultaneously recognize speech and eating behavioral sounds. The experimental results demonstrate that the proposed method achieved high detection accuracy in eating behavior recognition while maintaining robust speech recognition performance. Furthermore, the stepwise learning strategy employed in the two-stage fine-tuning process facilitated the effective integration of linguistic and eating behavioral features, thereby contributing to stable recognition performance across both domains.

In future work, we will further validate the practicality of the proposed method by collecting and evaluating data from real-environment data. Additionally, we aim to address the technical challenges associated with practical implementation, such as managing nonverbal sounds like laughter during meals and enhancing synthetic data generation methods to include more realistic overlap patterns. The findings of this study highlight new possibilities for speech recognition technology in daily health monitoring, offering significant potential for future advancements.

## Figures and Tables

**Figure 1 sensors-25-01544-f001:**
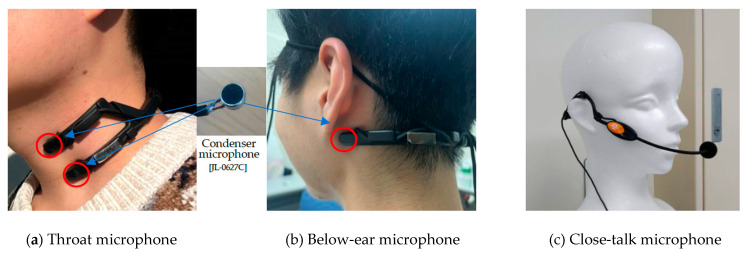
Skin-contact microphones and close-talk microphone. (**a**) Throat microphones are configured in two channels, positioned on the upper and lower parts of the throat. (**b**) Below-ear microphones are configured in two channels, positioned on the left and right below the ears, near the oropharynx. (**c**) Close-talk microphone.

**Figure 2 sensors-25-01544-f002:**
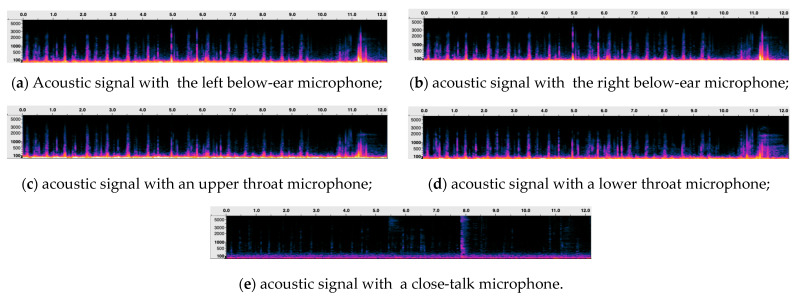
Examples of eating behavior sounds recorded using various skin-contact microphones. The spectrogram shows 15 chewing actions and 1 swallowing action. (**e**) represents a simultaneous recording using a close-talking microphone for comparison. It is evident that the close-talking microphone does not capture the eating behavior sounds. For illustration purposes, spectrograms of the recorded signals were generated using a short-time Fourier transform (STFT) in Audacity with the following parameters: 2048-sample Hann window, 50% overlap, and a Mel frequency scale spanning from 1 Hz to 11,025 Hz.

**Figure 3 sensors-25-01544-f003:**
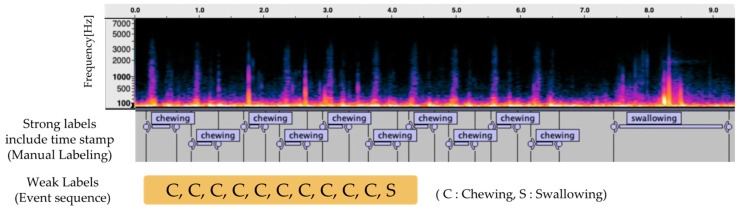
Example of annotations for below-ear microphone audio. This figure presents examples of strong and weak labels for eating behavior recognition. It consists of a spectrogram (top), strong labels with time information (middle), and weak labels as a sequence of events (bottom). Strong labels are used for evaluation, whereas weak labels are used for model training.

**Figure 4 sensors-25-01544-f004:**
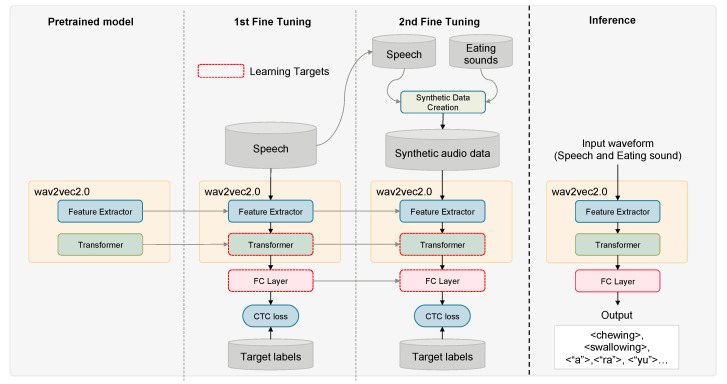
Overview of the simultaneous speech and eating behavior recognition model structure and learning methods. From left to right: the pre-trained model, the first stage of fine-tuning, the second stage of fine-tuning, and the inference process. The first-stage fine-tuning uses only speech data for training, whereas the second-stage fine-tuning uses synthetic data combining speech and eating behavior sounds. The red dashed lines indicate the modules where the parameters are updated during training.

**Figure 5 sensors-25-01544-f005:**
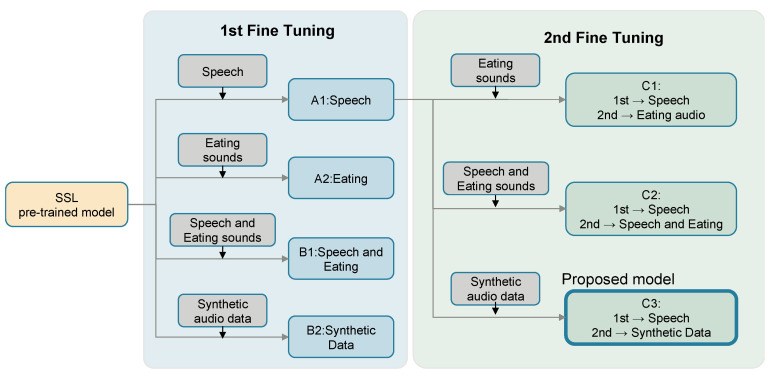
Overview of the comparison methods. The gray blocks represent the training data. Three types of training data were prepared: one where speech and eating sounds were treated independently (speech; eating audio), one where both were combined (speech and eating audio), and synthetic data (synthetic data). In the first stage of fine-tuning, four different methods were evaluated: A1: speech only, A2: eating sounds only, B1: independent processing of speech and eating sounds, and B2: synthetic data. In the second stage of fine-tuning, three methods were evaluated: C1: eating sounds only, C2: combination of speech and eating sounds, and C3: synthetic data. All methods are based on a common SSL pre-trained model.

**Figure 6 sensors-25-01544-f006:**
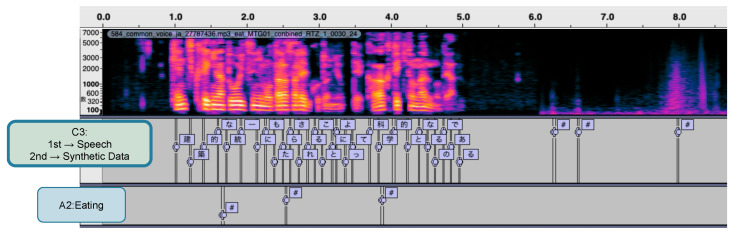
Comparison of recognition results between the proposed method (C3) and the baseline model (A2). Regarding the labels in the figure, ‘#’ indicates chewing events. The Japanese text represents the recognized speech content, which is the native language used in our experiments. The upper spectrogram shows the input audio, the middle shows the recognition results from the proposed method (C3), and the lower shows the results from the baseline model (A2).

**Table 1 sensors-25-01544-t001:** Character error rate (CER) and F1 score for each comparative method.

Training Data			CER ↓ [%]	F1 Score ↑	
	1st Fine-Tuning	2nd Fine-Tuning	Test: Speech	Event: Chewing (Allowance = 0.05 s)	Event: Swallowing (Allowance = 0.01 s)
A1	Speech (baseline)	-	15.19	-	-
A2	Eating (baseline [25])	-	-	0.905	0.963
B1	Speech and Eating ^2^	-	17.13	0.769	**0.961**
B2	Synthetic Data ^3^	-	16.38	0.835	0.654
C1	Speech	Eating	87.45	**0.903**	0.825
C2	Speech	Speech and Eating ^2^	**16.22**	0.243	0.514
C3 ^1^	Speech	Synthetic Data ^3^	**16.24**	**0.918**	**0.926**

^1^ Proposed method, ^2^ method treating eating sounds and speech as separate training datasets, ^3^ method using synthetic data combining eating sounds and speech as training data. Note: The bold values indicate the top two models whose results either match or exceed the single-task baseline performance.

## Data Availability

The original contributions presented in this study are included in the article. Further inquiries can be directed to the corresponding author.

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
