# Peer review of "Simultaneous Speech and Eating Behavior Recognition Using Data Augmentation and Two-Stage Fine-Tuning"

_sensors, 2025, doi:10.3390/s25051544_

Round 1

Reviewer 1 Report

Comments and Suggestions for Authors

Very interessting manuscript on Simultaneous Speech and Eating Behavior Recognition Using
Data Augmentation and Two-Stage Fine-Tuning.

In the introduction, authors might more clearly state what the differences to their previous publication (https://www.mdpi.com/1424-8220/21/10/3378) are.

Some other suggestions are given in the attached pdf file

Reviewer 2 Report

Comments and Suggestions for Authors

While the paper presents an interesting and potentially valuable approach, it has several critical weaknesses that need to be addressed before it can be considered for publication.

1. The paper mostly compares different versions of its own method instead of benchmarking against state-of-the-art models. How does this perform against other multitask learning models for speech and sound recognition? Without that, it’s hard to judge the real impact.
2. Some of the accuracy differences are really small (like CER 16.22% vs. 16.24%). Are these meaningful improvements? Running significance tests or showing confidence intervals would make the results more convincing.
3. The two-stage fine-tuning idea is interesting, but the explanation is too technical. A simpler breakdown of why this approach works better would help make it clearer.
4. Table 1 is packed with numbers, but a graph or visual comparison would make it much easier to see trends. Also, Figure 6 should highlight where the proposed method improves or struggles.
5. All tests are done in a lab, but real life is messy. What happens in a noisy restaurant? What if the microphone placement shifts? Right now, the results don’t show how well this would actually work in daily use.

Round 2

Reviewer 2 Report

Comments and Suggestions for Authors

accept